# Artificial lighting affects the landscape of fear in a widely distributed shorebird

Juho Jolkkonen [1], Kevin J. Gaston [2] & Jolyon Troscianko [3✉]

Fear influences almost all aspects of a prey species' behaviour, such as its foraging and movement, and has the potential to cause trophic cascades. The superior low-light vision of many predators means that perceived predation risk in prey is likely to be affected by light levels. The widespread and increasing intensity of artificial light at night is therefore likely to interfere with this nocturnal visual arms race with unknown behavioural and ecological consequences. Here we test how the fear of predation perceived by wintering Eurasian curlew foraging on tidal flats is influenced by lighting. We quantified flight initiation distance (FID) of individuals under varying levels of natural and artificial illumination. Our results demonstrate that FID is significantly and substantially reduced at low light levels and increases under higher intensity illumination, with artificial light sources having a greater influence than natural sources. Contrary to the sensory-limitation hypothesis, the curlews' unwillingness to take flight in low-light appears to reflect the risks posed by low-light flight, and a desire to remain on valuable foraging grounds. These findings demonstrate how artificial light can shape the landscape of fear, and how this interacts with optimal foraging decisions, and the costs of taking flight.

[1] Department of Biological and Environmental Science, University of Jyväskylä, Jyväskylä, Finland. [2] Environment & Sustainability Institute, University of Exeter, Penryn Campus, Penryn, Cornwall, UK. [3] Centre for Ecology & Conservation, University of Exeter, Penryn Campus, Penryn, Cornwall, UK. ✉email: jt@jolyon.co.uk

Perceived predation risk can have dramatic consequences for individual organisms, populations, or even whole ecosystems[1–3]. This is because the fear of predation can influence where prey species live and what they eat (landscape of fear theory, see e.g., ref. [4]) with the potential for causing cascading effects through the trophic levels. Vision is a critical sensory modality in predator-prey relationships, affecting animals' perceived predation risk (e.g., based on levels of cover by undergrowth[5,6]), their choice of microhabitat[7–9], and survival[10]. Perhaps the largest change to the global visual environment over recent decades has been the introduction of anthropogenic artificial light at night (hereafter 'ALAN') from homes, businesses and infrastructure[11,12]. This now extends over a high proportion of the landscape[13,14] and has been found to have a wide array of biological impacts on individual species[15]. Its effects on interspecific relations are likely to be widespread, although these have been little explored (but see e.g., refs. [16,17]). In particular, changes in nighttime lighting caused by artificial sources are predicted to alter the abilities of prey species to detect predators, and vice versa, with the potential for a dramatic reshaping of the landscape of fear.

Wintering shorebirds provide a valuable study system to investigate such effects. The activity patterns of many of these species are largely governed by the tidal cycle's influence on food availability, rather than the day/night cycle[18,19], forcing them to forage during nocturnal low tide periods to balance their daily energy budgets[20,21]. Shorebird vision has therefore evolved to combine nocturnal and diurnal panoramic predator vigilance with other roles, whereas their nocturnal predators typically have eyes better suited to low-light performance, with rod-dominated retinae, narrower visual fields, higher visual acuity, and more powerful optics[22,23]. Low light levels are therefore likely to shift the visual advantage to predators in this system. Finally, the tidal flats on which much wader foraging activity takes place is increasingly lit artificially as a result of coastal development, with an absence of topographic features to obstruct the path of horizontal light emissions[24].

Waders and gulls have been observed to increase their nocturnal foraging activity in the vicinity of artificial light sources[25–28]. Moreover, waders switch from tactile to visual foraging under either artificial illumination or bright moonlight[27,29]. While these results suggest that higher artificial and natural illumination improve birds' visual perception of their prey at night, we do not know how it affects their own perceived predation risk. Here we use a standardised assay for inferring perceived predation risk in birds, the flight initiation distance (hereafter 'FID')—the distance at which individuals take flight when approached by a potential predator (e.g., refs. [9,30,31]). Different escape distances have relative costs and benefits; fleeing is energetically costly and disrupts foraging behaviour, but failure in risk assessments can be fatal[32]. Despite considerable previous work testing FID of various avian species in different habitats, surprisingly little is known about how light levels affect birds' perceived predation risk. Grey partridges *Perdix perdix* have, for example, shorter FID at night, and this is thought to be caused by the limitations of their low-light vision[33]. However, the nocturnal light environment is highly variable, being dependent on the lunar cycle, artificial light, and atmospheric conditions[11]. The intensity of ALAN widely exceeds the brightness of moonlight[11], potentially acting in many areas as one of the main determinants of foraging waders' perceived predation risk and the landscape of fear at night.

In this study, we quantified the FID of wintering Eurasian curlew *Numenius arquata* ('curlew' hereafter), foraging on tidal flats both in the daytime and at night. In the recent IUCN Red list assessment in 2017, the curlew was listed as Near Threatened[34], and the species has been under high conservation effort in the UK due to rapid decline in its population size[35,36]. Therefore, a better understanding of the anthropogenic effects on their foraging behaviour and foraging site selection may provide important tools for the species conservation. We used an approaching human to simulate a potential threat (a standard technique; e.g., refs. [9,30,31]), and a thermal imaging scope to observe the curlew's escape behaviour (for full details, see Methods). "Fear" is a mental state that is impossible to quantify in non-humans, however FID provides a good assay because it forces the bird to trade off the fear induced by an approaching threat, against the fear (costs and risks) associated with taking flight. To investigate whether the light environment affects their perceived risk of predation, we used a high-sensitivity spectroradiometer to measure the varying levels of artificial and natural illumination within the study area at night. Nocturnal light intensity might be expected to affect FIDs through four (non-exclusive) drivers, by: (i) affecting the curlews' ability to detect approaching predators ("sensory limitation theory"[33]); (ii) hesitancy in taking flight due to the increased collision risk in darker conditions ("flight avoidance theory"[37]); (iii) decreasing the perceived predation risk as the curlew feel more hidden and safe in darkness ("cloaked in darkness theory", akin to landscape of fear examples where visual obstruction decreases perceived predation risk[38,39]; or (iv) increasing the curlews' perceived risk in darkness due to their limited visual abilities compared to those of their nocturnal predators ("fear of darkness theory", similar to instances where visual obstruction increases the perceived risk; e.g., ref. [6]). The first three hypotheses would predict increasing FIDs with light intensity, while the last predicts the reverse under very low light, fleeing at the first sign of danger. To test these hypotheses, we modelled curlew FID ($n = 86$) against two light intensity measures (i.e., horizontal ALAN and downwelling light) together with the curlew's latitude and longitude as covariates to control for spatial autocorrelation in escape behaviour. Additional covariates in the model were flock size, time since/until the nearest low tide, tide height at low tide, date (day of the year, to control for habituation over time) and temperature. See supplementary data file for R script with analysis code and detailed model output.

## Results and discussion

**Light intensity affects the perceived risk of predation.** Higher illumination from both artificial light sources and downwelling light from the sky was found to increase nocturnal FIDs substantially (Fig. 1a, b; Table 1 & S1), in line with the sensory limitation, flight avoidance, and cloaked in darkness hypotheses. Furthermore, nocturnal FIDs were shorter than daytime FIDs (Fig S2a) but FIDs increased on moonlit nights (Fig S2a), and individuals foraging in larger flocks had shorter FIDs (Table 1, Fig S2b), which supports the theory of the larger group size's protective effect reducing the perceived predation risk[40,41]. Curlew also had shorter FIDs nearer low tide (Fig. 1c). Previous research has emphasised how cyclic availability of food determines waders' foraging activity[18,19], so the disruption caused by fleeing would likely be most costly during the lowest stage of the low tide period when the availability of food resources is highest. Additionally, FIDs were marginally shorter (although not significantly, $p = 0.098$) during spring tides, when larger foraging grounds are exposed at low tide. These tidal effects support the hypothesis that FID is linked to optimal foraging behaviour, where the curlew allow potential predators to come slightly closer when they are foraging on more valuable food resources[42–45]. Temperature, date, latitude, and longitude did not improve the model fit and were dropped from the model that best explained FID (see Table S2 for the full factorial model).

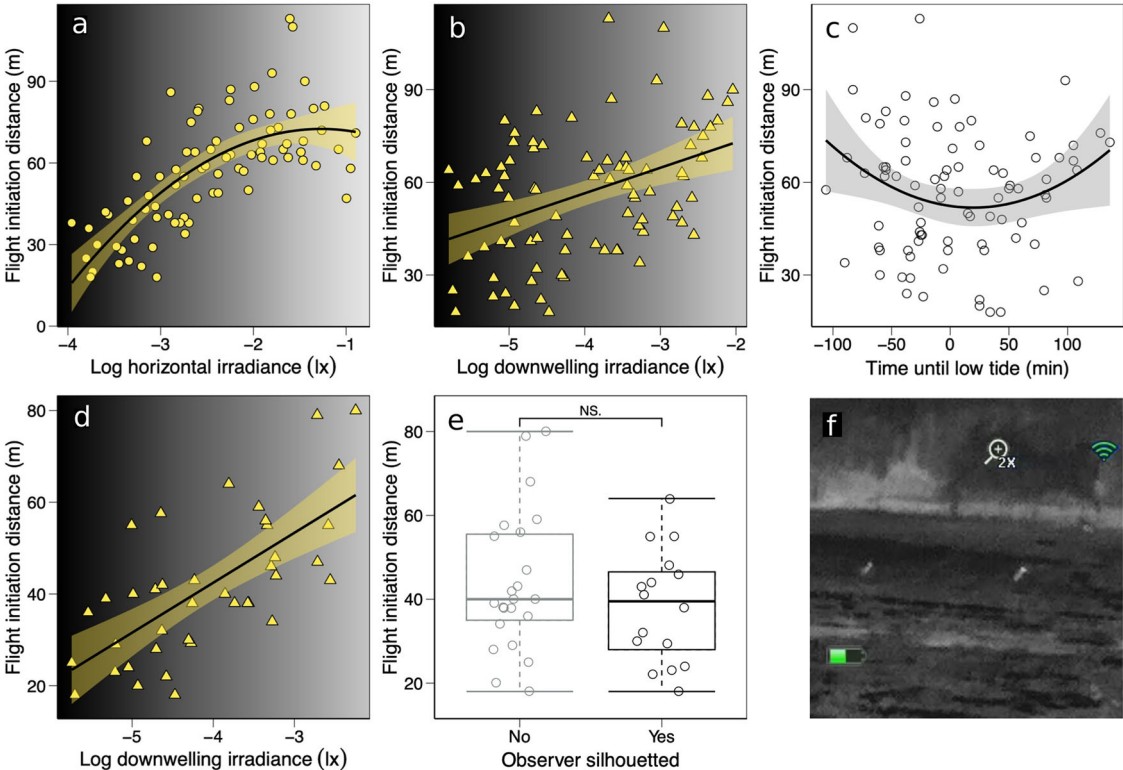

**Fig. 1 Nocturnal light environment affects curlew FID.** Regression plots show how the higher intensity of horizontal ALAN **a** and downwelling light **b** increased curlew FIDs. **c** FIDs were shorter around the lowest stage of the low tide period ($n = 86$ in plots **a–c**). When considering only the measurements taken in the 'creek' where the levels of horizontal ALAN were low, FID was strongly influenced by downwelling light **d**. Foraging curlew were approached from different directions (i.e., being silhouetted against the horizontal artificial lights or a dark woodland), but this did not have a significant effect on FID **e**. **f** Two curlews observed through the thermal camera at night ($n = 39$ in plots **d**, **e**).

**Table 1 Results of the best linear model explaining curlew's FID at night.**

| Variable | Estimate | SE | t | p |
|---|---|---|---|---|
| (Intercept) | 56.63 | 1.29 | 43.79 | **<0.001** |
| Horizontal ALAN, linear | 103.77 | 14.48 | 7.17 | **<0.001** |
| Horizontal ALAN, polynomial | −19.59 | 13.14 | −1.49 | 0.140 |
| Downwelling light, linear | 52.73 | 13.32 | 3.96 | **<0.001** |
| Downwelling light, polynomial | 28.21 | 12.14 | 2.32 | **0.023** |
| Flock size | −4.44 | 1.72 | −2.59 | **0.011** |
| Time until low tide, linear | −11.04 | 12.91 | −0.85 | 0.396 |
| Time until low tide, polynomial | 44.11 | 12.60 | 3.50 | **<0.001** |
| Low tide height | 2.46 | 1.47 | 1.68 | 0.098 |

Bold values indicate statistical significance $p < 0.05$.
Horizontal ALAN, downwelling light (linear and second-order polynomial), flock size and time until low tide were the significant predictors of the FID in the best, simplified model.

**Sensory limit vs. flight avoidance**. Finding a safe area to land at night is likely to be more difficult under lower light levels, causing birds to stay on the wing for longer and use more energy. Night-time flight is also known to increase the risk of potentially fatal collisions[37]; during the course of this study, we observed a mallard *Anas platyrhynchos* colliding with the rigging of a moored sailing boat. Alternatively, our finding that nocturnal FIDs increase with light levels could simply reflect an inability in the curlew to see the approaching experimenter[33]. We can potentially distinguish between the "flight avoidance" and "sensory limitation" theories if we assume that silhouetting from horizontal ALAN affects the ability of the curlew to detect the approaching experimenter, i.e., when the experimenter or a predator is between the curlew and an artificial light source their moving silhouette could either reveal their approach, or could conceal their approach if the ALAN causes

glare or photobleaching. Indeed, predators are known to take advantage of light direction in the daytime, such as raptors attacking from the sun[46] and prey species may respond to that by avoiding foraging in direct sunlight[47,48].

Assuming that a moving silhouette increases the detection distance of the approaching human, we would predict longer FIDs from backlit, silhouetted observers. Alternatively, we could assume that a backlit observer is more difficult for curlew to detect (*sensu* ref. [46]), decreasing the detection distance and FID. As the study site's main sources of ALAN were from the south (see Fig S1), only individuals foraging on the northern side of tidal flats (hereafter 'creek') could be approached with a silhouetted profile. Therefore, we included only the FID measurements taken in this area when analysing the silhouette effect ($n = 39$). We followed the previous simplified model (Table 1) but included backlight as one of the

explanatory variables to test whether approaching while backlit increased FID. Contrary to our predictions, we found no significant effect for the approaching predator being backlit (Fig. 1e, Table S3). Instead, the positive effect of downwelling light (i.e., natural moonlight) on FID became more pronounced (Fig. 1d) in this area, where the lower intensity of horizontal ALAN (see Fig. 1) did not have any significant effect on FID (Table S3). Neither flock size, nor time until low tide had a significant effect on FIDs (Table S3), which could imply that an unwillingness to take flight and/or land in low-light governs FID decision making, although the null may also reflect a modest sample size ($n = 39$).

Nocturnal FIDs in curlew were found to increase substantially under higher levels of both artificial and natural light. Additional lines of evidence suggest that this increase in FIDs is not limited by the curlews' ability to detect an approaching threat, but rather an unwillingness to take flight under lower light levels and a willingness to stay put on valuable foraging resources. First, consistent with predator vigilance predictions, FID was reduced with larger group sizes. The opposite effect would be expected if the relationship between FID and nocturnal light intensity was exclusively driven by detection distance at low light levels (more eyes should detect approaching threats sooner). Second, contrary to the silhouette hypothesis, backlit observers neither increased nor decreased FID (we assume one of these conditions is likely to be more salient than the other), implying detection limits alone are unlikely to explain the effect. Nevertheless, we cannot rule out the detection-limited theory when the visual detection is more difficult, such as under light levels lower than those experienced in our field site or when a natural, less conspicuous predator such as the red fox *Vulpes vulpes* slowly approaches the avian prey from a low angle. For example, previous research has monitored reduced activity in jackals during the full moon period, which has been suggested to be due to impaired hunting efficiency caused by the prey species' improved visual detection[49]. Red foxes were observed on the tidal flats regularly throughout the season and caused the curlew to take flight, therefore representing a clear threat for the curlew foraging at night (Video S1 & S2).

Our results also show that—consistent with optimal foraging theory[50,51]—FIDs were shorter when tides were at their lowest. This likely reflects the curlew's desire to remain at higher value foraging locations and shows how cost/benefit trade-offs of foraging affect FID, and we would not have detected this effect if FIDs were exclusively dependent on sensory limitations. Previous studies have found that foraging rodents are more susceptible to predation by owls under moonlight illumination (when owls also increase their activity), and in response may reduce their foraging activity or seek protection from more complex habitats and shadows[38,39]. On the open tidal flats, foraging waders lack the physical and visual protection offered by vegetation, potentially emphasizing the importance of the light environment on their susceptibility to predation. Therefore, contrary to the "fear of darkness" theory, the curlew could perceive a lower risk of predation when foraging in darker areas (i.e., feel safer in the darker areas and therefore take flight later; "cloaked in darkness theory") and avoid any unnecessary movements, whereas under higher illumination, perceived risk is higher, and the curlew are more sensitive to escape. Nocturnal foraging has been suggested to protect shorebirds against their diurnal predators[21], and different anti-predatory tactics may be more efficient against nocturnal predators. For example, whereas an approaching peregrine falcon *Falco peregrinus* causes panic in flocks of dunlin *Calidris alpina* in the daytime, dunlin avoid nocturnal predators (owls) by remaining motionless and silent at night[52]. Our results of reduced FIDs in lower light levels may suggest a similar behavioural change in the curlew but further experiments would be required to disentangle this effect from the flight avoidance theory.

Taken together, the above effects demonstrate how light intensity (artificial and natural) influences optimal foraging decisions, perceived predation risk and the costs of taking flight. While the benefits of ALAN in waders' visual foraging at night have been observed[27,29] and some species use foraging areas illuminated by street/floodlights more than areas without artificial illumination (however, see ref. [29]), how ALAN affects the survival of wintering waders remains unknown. Our results suggest ALAN may benefit the waders in reducing the costs associated with taking flight and could be predicted to reduce the optimal light conditions for the predator approaching the prey (naturally determined by the moon phases; ref. [49]). We propose that future efforts tracking wader movements could further reveal the influence of ALAN in local foraging site selection with varying pressure from different nocturnal predators (mammals and owls), potentially shifting the optimal foraging strategies and the landscape of fear.

## Methods

**Ethical statement**. Ethical permission for the project was granted by the University of Exeter (CLES Cornwall Ethics Committee, ID 493710).

**Study area**. We collected data on the tidal mudflats of the Penryn River (centred on 50°10′05.4″N, 5°05′29.7″W), Cornwall, United Kingdom. The fieldwork was conducted from mid-October until mid-December in 2021 during the low tide period (water level below 2.5 m, tidal variation on site between ca 0.2–6 m), across 25 nights and 10 days. In total, FID was measured from 86 curlew at night and 27 individuals in the daytime. No other wader species were observed in the vicinity of the tested curlew individuals, minimising the potential interspecific effects in cooperative alarming and fleeing. Experiments were not conducted in the rain to minimise any weather-dependent variation in their behaviour and reduce potential harm to the birds caused by the study. Dawn and dusk were also avoided (i.e., not within 1 h before or after sunrise or sunset) due to rapid change in light levels. Temperature varied between 4 °C and 15 °C at night and 8 °C and 15 °C in the daytime during the study (weather data obtained from https://www.foreca.com/102640413/Penryn-Cornwall-United-Kingdom). The fieldwork was not conducted on windy nights to control for any acoustic cues of the observer and possible avoidance of taking flight in heavy wind. Other recorded environmental variables were the time from the FID experiment to the lowest stage of the low tide period (in minutes), and the low tide height (UK Hydrographic Office, https://www.bbc.com/weather/coast-and-sea/tide-tables/10/5).

**FID experiment**. We measured the flight initiation distance (FID) of foraging curlew in their wintering area by following a common protocol (e.g., refs. [9,30,31]) where a human observer approaches the target individual until it flushes. To find the birds and follow their behaviour when approached, we used a thermal imaging scope (IR510 Nano N1 Wifi, Guide Sensmart, Germany, Fig. 1f) at night and binoculars during the day. Highly vigilant, alarming, or roosting individuals were not approached. Furthermore, birds that were observed closer than 75 m were not approached to avoid any variation in FID caused by varying starting distances (i.e., a distance between the bird and observer when starting the approach[53]). Starting distance was not found to have any effect on FID (linear regression between the starting distance and FID: $F_{1,37} = 0.173$, $p = 0.680$). All the experiments were conducted by the same person (J. J.), wearing similar, brown-coloured clothes to control for potential observer-based variation. The foraging curlew was approached directly at a slow, consistent walking pace until the bird took flight. At that moment, the observer's location was recorded on GPS (OsmAnd application, Samsung Galaxy S10 + ). If the curlew were foraging in flocks, one of the birds was randomly selected for the experiment. Flock size was recorded as the number of individuals aggregated in a group, foraging within 10 m of the other individuals. Next, the location of the flushed bird was recorded on the GPS, and FID was later measured using the recorded coordinates of the observer and bird.

**Measuring nocturnal light levels**. Immediately after the bird had flushed, the nocturnal light environment was measured from the flushing point using an OSpRad high-sensitivity spectroradiometer[54] ($n = 86$, points marked on the heat maps in Fig. S1A and S1B). OSpRad spectroradiometers are built around Hamamatsu C12880MA chips, and are driven by Arduino Nano microprocessors. Their construction, calibration, and processing have been released open-source[54]. OSpRads use a servo-driven shutter wheel to automatically record irradiance through a cosine corrector made from four layers of 0.1 mm thick virgin PTFE. The shutter wheel then blocks the incident light for the chip's dark measurements to be made and subtracted from the light measurement. The spectroradiometer was connected to a smartphone (Wileyfox Spark X) through a USB interface, and custom-written software on the phone recorded the measurements. The system automatically took three measurements with integration times of up to 30 s.

The sensitivity of the system was calibrated against a NIST-traceable Jeti Specbos 1211-UV-2 spectroradiometer using a stable halogen-xenon light source. Artificial lights are ultimately deployed for the benefit of human visual systems, so irradiance measurements ($W\,m^{-2}\,nm^{-2}$) were converted to lux ($lm\,m^{-2}$) using the CIE Y (2006) sensitivity curve. With the spectrometer, we first quantified the intensity of direct, horizontal artificial light by measuring irradiance at 40 cm height, facing the brightest artificial light sources (e.g., streetlights and warehouses). Then we measured irradiance pointing directly upwards to quantify the downwelling light from the sky (i.e., including moonlight, light reflecting from clouds, skyglow). These measurements of the horizontal ALAN and the downwelling light were independent and did not correlate (linear regression between the downwelling light and horizontal ALAN: $F_{1,84} = 1.26$, $p = 0.265$).

**Statistics and reproducibility**. An unequal variance Welch's t-test was used to calculate the difference between the daytime/nighttime FIDs and moonlight/moonless night FIDs (Fig S2). We modelled how FID was affected by direct artificial illumination ('Horizontal irradiance') and downwelling light ('downwelling irradiance') from the sky. Latitude and longitude of the measured individual were included in the model to control for any site-dependent variation in curlew escape behaviour. Time since/until the nearest low tide, and low tide height were included in the model to explain any tide-dependent variation in FID. Finally, date and temperature were included in the model to control for any seasonal change and to control for any learning/habituation effects through repeated encounters or temperature-dependent variation in curlew's response to an approaching predator. The curlew population was unmarked (although even marked individuals would not be identifiable at night), and we estimate that the population was around 50–70 individuals over the course of the study. As such, there is likely to be a small, yet unknown level of repeat measurement of individuals in our statistics. We attempted to control for repeated measures by sampling as widely as possible spatially and temporally, and by including date and GPS coordinates of the measured individual as covariates. We tested a full factorial model (Table S2) and then reduced non-significant independent variables that did not contribute to the model fit using the drop1 command. The stepwise model simplification was continued until the final model with the lowest Akaike information criteria (AIC) value was gained. The most appropriate model error structure for each responsible variable was checked using QQ-normality plots and residuals vs. predicted plots from the DHARMa package[55]. Measurements of direct ALAN and downwelling light were log-transformed to meet model assumptions and all the other explanatory variables were standardised by centring and scaling, using the scale-function in R.

As the artificial light sources were situated on the south bank of the river (tidal flats, Fig. S1a), the study system has strong collinearity between horizontal ALAN and latitude within the study site (VIF-function: $GVIF_{ALAN} = 6.94$, $GVIF_{Latitude} = 5.82$). When stepwise model simplification was repeated including the higher-order polynomial terms of latitude as an explanatory variable (i.e., poly(latitude,3)), the third-order polynomial term was found to be a significant predictor of the FID. Nevertheless, including the second and third-order polynomial terms in the model did not qualitatively change the results; all the other significant predictors remained significant.

When testing the silhouetting effect, backlight (binomial variable) was included in the simplified model. However, as the study site's main sources of ALAN were from the south (Fig S1a), only individuals foraging in the creek area could be approached with a silhouetted profile from the south. Therefore, the dataset used for testing the silhouette effect consisted of measurements taken to the north of latitude 50.16873 °N (dashed line in Fig. S1a and S1b) and we analysed these data separately. All the statistical analyses were performed in R version 4.1.2[56].

**Reporting summary**. Further information on research design is available in the Nature Portfolio Reporting Summary linked to this article.

## Data availability
Data and analysis R script are included as supplementary data (supplementary data.zip).

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

## Acknowledgements

J.J. was funded by an Erasmus+ Student Work Placement; J.T. was funded by NERC Fellowship NE/P018084/1; K.J.G. was supported by NERC grant NE/V000497/1; J.T. and K.J.G. were supported by NERC grant NE/W006359/1.

## Author contributions

J.J. and J.T. designed the field experiments and analysed the data. J.J. conducted the field experiments. J.J., J.T., and K.G. wrote the paper.

## Competing interests

The authors declare no competing interests.
