## [Peer Review File · Communications Biology]

Reviewers' comments:

Reviewer #1 (Remarks to the Author):

Reviewer comments to the manuscript 'Hereby my comments to the manuscript: 'Artificial lighting affects the landscape of fear in foraging shorebirds' by Juho Jolkkonen, Kevin J. Gaston and Jolyon Troscianko

The authors report the outcome of a study on the Flight Initiation Distance (FID) of curlews foraging on mudflats in the Penryn River, and how artificial light at night affects this parameter. FID is often taken as a proxy for (a combination of) vigilance, ability to sense danger and perceived predation risk. The main outcome of the study is that the distance birds go on the wings substantially increases with increased light levels.

Knowing how the response of organisms to (potential) predators depends on the presence and intensity of artificial light at night is highly relevant as this changes the interaction of species and hence makes effects cascade within the food web. I have some concerns and comments, mostly related to the ubiquity of the results and confounding effects.

The study is non-experimental and essentially done at just one location and the lighting and foraging conditions may be relatively specific for the Penryn river estuary. Testing the hypothesis at many comparable locations is however logistically challenging. The impact of the work, and the assertion of the title would benefit if the ubiquity is better motivated.

The study is potentially burdened with confounding effects, hence the work would benefit from a better explanation on how:

- the flock size relates to light intensity – this may explain the shorter FIDs with larger flocks?
- the (line 230) vigilance relates to light intensity, and subsequently the likelihood of inclusion of the study? How does the alarming (and the response to ALAN) of other species relate to the response of curlews?
- The predator density may vary within the site
- The distance to shore relates to light intensity - the distance foxes may need to walk to get to their prey may determine predation risk, and hence explain the response to light?

Line 231 – the solution to avoid variation in FID seems to be highly arbitrary, as I can foresee that other differences between larger start distances in approaching can as well cause differences – variation – in responses. Can the authors provide figures on how other start approaching distances relate to the FID's observed?

Line 239 – how did flock birds respond? Would they not leave (as commonly observed) as flocks, and hence selecting an individual would not be necessary – they would all leave with the first bird scared enough to leave (see also remark on effect of flocks above)?

Line 241 – FID values can range to up to 80 m, how was ensured the flushed bird location was assessed accurately?

Minor comments

The evaluation of the four hypothesis is a bit of a storytelling as these are not directly tested and only circumstantial evidence yields a bit more or less support for some of these, for example the assumption that because of the reduced FID in larger flocks, the increased sensing of predators is not the main driver, but instead the increased FID may be the ability to take flight because of reduced collision risk. This is not much of a problem, but yields a relatively lengthy manuscript.

Line 74 – driver (i): one could as well expect birds to fly later because they are better able to detect predators?

Line 94 – flight avoidance reads counter intuitive – in fact they fly earlier, but what is meant is the flight avoidance collision theory – hence maybe better write ‘flight avoidance hypothesis’

Line 114 – this is really far-fetched – a different species and totally anecdotal; it does not much strengthen the story

Line 116 – why would FIDs increase with birds less able to see the experimenter? They would then decrease?

Reviewer #2 (Remarks to the Author):

Brief summary of the manuscript

This manuscript aims to investigate whether artificial light at night (ALAN) affects prey behaviour (in this case Eurasian curlew). A range of hypotheses are presented for how birds may respond to light (or lack of) at night; sensory limitation, flight avoidance, cloaked in darkness (perceived safer) or fear of darkness (perceived riskier). Flight initiation distance in Curlew increased with ALAN, in line with the flight avoidance hypothesis (although other hypotheses cannot be entirely ruled out).

Overall impression of the work

Understanding the effects of anthropogenic effects such as light and noise is a very important area of research given the problems many wild animal populations face. Much has been done with noise in birds, but less with light, so I think this manuscript presents an original contribution to the field. Furthermore, I liked that it was especially grounded in theory, with good consideration of a range of hypotheses, well-linked to the established literature.

I thought the standard of writing was generally high, with a clear narrative, that made it easy to understand what was done and why. My main criticism of the narrative is the lack of conservation emphasis. I appreciate there is a tight word limit and it is important to cover the behavioural theory but I do think it needs to be considered and emphasised more what the challenges are for shorebirds – there is a lot of conflict with development and recreation along coasts (of which ALAN is part of) and the Eurasian Curlew is particularly interesting since it has massively declined in the UK and is likely to go extinct in some areas (e.g. Wales). I think where it is argued shorebirds are a good model system in the introduction you could also introduce the conservation context without distracting from the behavioural narrative too much. I don't think your study needs to prove that ALAN is “bad” (indeed your study suggests it's more complex than that!) to be able to set things in a conservation context and attract a wide readership.

The study design is generally good. It is not experimental so there are potential confounds to be considered (similar to research along urban-rural gradients) e.g. other variables besides light could vary across the study site. These could perhaps be considered a bit more – there is consideration of environmental confounds such as temperature but less discussion of what might vary across the site along the same gradient as light (i.e. other anthropogenic effects like disturbance or noise).

However, I do think we need a combination of observational field studies such as this and smaller scale experiments (it would be quite hard to experimentally manipulate this particular study system),

so I take no major issues with having an observational study per se.

I am not an expert at measuring light, but the methods here seemed generally detailed and robust. What was less clear was whether there were likely to be repeated measures on the same Curlew (i.e. potential for pseudoreplication). I appreciate that the birds were unlikely colour ringed (or possible to read rings of anyway at night!) but some estimate of the sample size of the population size and therefore the likelihood of re-sampling would be useful for transparency over pseudoreplication possibilities (even if there is no data for this particular bay, there should be populations within the literature that give an estimate of general population size/density).

Finally, I wasn't too sure whether there is potential for observer bias since the observer was estimating the birds location on GPS – how accurate is this generally? I appreciate that using a laser rangefinder might be difficult at night and many studies likely use an estimation technique such as this, but did you measure your accuracy and check it didn't vary according to light levels? I would assume that I would be worse at estimating distance in the dark versus in the light, particularly in an environment where there aren't very many landmarks to relate a birds position to in a coastal context. Since most FID studies are done in daylight and don't look at a light gradient, I think that we can't necessarily assume the same accuracy using similar methods and it needs a bit more justification.

Overall though, I expect most of these comments can be addressed with minor revisions to the manuscript and I think this would be a valuable contribution to the published literature.

Specific comments

Lines 67-89. I thought there were some details here that were more appropriate to have in the methods or results section (e.g. references to figures and tables on line 73, detail about modelling from lines 84-89, reference to R code on line 89). I would cut these, as they repeat info in other sections and that would create more word count that could be used to introduce context on conservation status of Curlew.

Line 217. What about high winds? Wind speed can affect vigilance as the wind acts to mask noise so this could interact with FID.

Line 220. I found it unclear here how you obtained weather data. The website you link to is just a weather forecast website. A weather forecast is different to measuring actual temperature at your field site or getting data from a nearby weather station.

Line 222. www.bbc.com is a very vague web address – it's just the entire bbc site. Provide a specific link for where you got the information from.

Line 241. See point above – minimal on how bird location was recorded and the accuracy of this and whether light affects accuracy of estimation.

Line 258. I assume that horizontal light was measured at "bird height" not chest height of the observer? Needs to be stated what height above ground the horizontal measures were taken.

Line 273. Related to my point on line 220, it's unclear whether you actually measured temperature and how or whether you were using weather forecast data.

Line 437. I liked that you reported your full model and that the results are similar whether you use a step-wise approach or full model – gave me extra confidence that the effect is large enough to be robust to what statistical methods you use.

Reviewer #3 (Remarks to the Author):

In this article, the authors assess the FID of the Eurasian curlew in relation to light levels (moon and artificial light). They use FID to humans as a proxy for the fear felt by the birds. I like reading the paper and I think that the scientific question is interesting and timely, but I felt that it was too pretentious.

Given this study was conducted with a single species, it is too pretentious to say that artificial light affects the landscape of fear in shorebirds (just one studied!). Furthermore, and more importantly, the study area is quite limited in extension (around 600 m long). So, I bet that the number of curlews is also limited. Given that wintering population size is not given, I assume the authors have sampled many birds multiple times. Also some clusters of focal birds are observed in Fig S1. Can they be the same individuals? This is essential to answer such a question, as it affects statistical analyses (random factors) and to extrapolate to all shorebirds as the title proposes. I also miss a wider discussion on how artificial light can modify the landscape of fear of other birds (for example, other non-prey birds such as birds of prey or owls).

FID studies and the landscape of fear are not the same. It is true that as a first approximation it could be useful, but a better description should be done.

Line 215: what water level is this one? maximum tidal level? what is the tidal variation in depth?

Line 216: It must be reworded. Do you know if they are 86 and 27 different individuals? Are they ringed? How did you identify them?

Lines 266: Are predictors standardized prior to modeling?

We would like to thank the reviewers for their constructive and thorough suggestions. The reviewers all highlight the value of the study, while appreciating the limits of observational fieldwork such as this. We have made a number of changes to the manuscript which are largely focussed on increasing clarity and providing specific detail. Below we respond to each reviewer's points in *blue italics*.

Reviewers' comments:

Reviewer #1 (Remarks to the Author):

The authors report the outcome of a study on the Flight Initiation Distance (FID) of curlews foraging on mudflats in the Penryn River, and how artificial light at night affects this parameter. FID is often taken as a proxy for (a combination of) vigilance, ability to sense danger and perceived predation risk. The main outcome of the study is that the distance birds go on the wings substantially increases with increased light levels.

Knowing how the response of organisms to (potential) predators depends on the presence and intensity of artificial light at night is highly relevant as this changes the interaction of species and hence makes effects cascade within the food web.

We are glad that the referee agrees the work has the potential to highlight important ecological effects.

I have some concerns and comments, mostly related to the ubiquity of the results and confounding effects. The study is non-experimental and essentially done at just one location and the lighting and foraging conditions may be relatively specific for the Penryn river estuary. Testing the hypothesis at many comparable locations is however logistically challenging. The impact of the work, and the assertion of the title would benefit if the ubiquity is better motivated.

While we accept the scope of our study is inevitably limited, we also believe that our paper is the first to point out the potential wide ranging behavioural impacts of ALAN on fear, and these principles could be widespread in other species. However, we renamed the article: "artificial light affects landscape of fear in a widely distributed shorebird", according to the suggestions by reviewers 1 & 3.

The study is potentially burdened with confounding effects, hence the work would benefit from a better explanation on how:

- the flock size relates to light intensity – this may explain the shorter FIDs with larger flocks?

Flock size was retained in the simplified model, showing its importance in explaining FIDs. Possible autocorrelation between flock size, light intensity and FIDs has been accounted for in the full model. Therefore, the reviewer is correct that flock size is a factor affecting FIDs, but light levels have a much more substantial effect in the model that accounts for variance in flock size. e.g. see table 1.

- the (line 230) vigilance relates to light intensity, and subsequently the likelihood of inclusion of the study? How does the alarming (and the response to ALAN) of other species relate to the response of curlews?

In our study site there were no mixed flocks of waders. Indeed, other wader species were very rare – during the study, we observed only a few redshanks at night foraging further away from the curlew(s). We have amended the manuscript line 207 to state "No other wader species were observed in the vicinity of the tested curlew individuals, minimising the potential interspecific effects in cooperative alarming and fleeing".

- The predator density may vary within the site

This may well be the case, although it would be extremely difficult to quantify in almost any natural site. We observed foxes traversing the creek opening on several nights, but we were not able to trace their movements.

- The distance to shore relates to light intensity - the distance foxes may need to walk to get to their prey may

determine predation risk, and hence explain the response to light?

This could in principle be true, however our analysis broke the study area down into two sites, and we observed the same effects in the creek-only dataset, where natural light alone affected the FID (in the location closest to presumed fox-related predation risk). Moreover, the effect we observed goes directly against the predicted effects described. E.g. the light intensity was lowest in the creek (furthest from the shore, and closest to presumed fox-related risk), however in this location FIDs were consistently lower.

Line 231 – the solution to avoid variation in FID seems to be highly arbitrary, as I can foresee that other differences between larger start distances in approaching can as well cause differences – variation – in responses. Can the authors provide figures on how other start approaching distances relate to the FID's observed?

We tested for the potential of this effect in a subset of our data where start locations were measured. We found no significant effect of start distance on FID, see R analysis code line 840, LM: $p=0.680$.

Line 239 – how did flock birds respond? Would they not leave (as commonly observed) as flocks, and hence selecting an individual would not be necessary – they would all leave with the first bird scared enough to leave (see also remark on effect of flocks above)?

It would not have been possible systematically to quantify the behaviour of the whole flock at night (through a thermal imaging scope), and our study was only concerned with quantifying risk from an individuals' perspective, so we randomly selected a single individual within the flock to observe.

Line 241 – FID values can range to up to 80 m, how was ensured the flushed bird location was assessed accurately?

Location could be ascertained from clear visual landmarks (ambient light (i.e. ALAN and moonlight), and/or the thermal camera revealed patterns in the mudflats composed of algae and rivulets etc. with varying temperatures). Under higher artificial illumination, these patterns and the tested curlew could sometimes be observed even without the thermal camera. Locations could also often be cross-verified through footprints, which often revealed individual steps, and the exact point of take-off (due to deeper, terminal footprints).

Minor comments

The evaluation of the four hypothesis is a bit of a storytelling as these are not directly tested and only circumstantial evidence yields a bit more or less support for some of these, for example the assumption that because of the reduced FID in larger flocks, the increased sensing of predators is not the main driver, but instead the increased FID may be the ability to take flight because of reduced collision risk. This is not much of a problem, but yields a relatively lengthy manuscript.

To our knowledge these hypotheses have not been spelled out in previous work (e.g. they focus solely on foraging or predation risk alone). We believe that a strength of our study is that we clearly synthesize a number of likely hypotheses, and test them in a natural system. Indeed reviewer 2 below praises our approach.

Line 74 – driver (i): one could as well expect birds to fly later because they are better able to detect predators?

Indeed, fear of predators might increase when the detection of predators is limited, for example, due to low light levels. Thus, one could predict that under higher illumination FID increases as the prey is better able to detect the approaching predator. This prediction is included in our hypothesis no 4, "fear of the dark", where we expect that under very low light levels, the curlew would flee at further distances due to increased fear of not being able to detect and follow the approaching threat.

Line 94 – flight avoidance reads counter intuitive – in fact they fly earlier, but what is meant is the flight avoidance collision theory – hence maybe better write 'flight avoidance hypothesis'

This sentence is part of a list where all of the theories listed are "hypotheses". We could change this to read: "[...]

in line with the sensory limitation hypothesis, flight avoidance hypothesis, and cloaked in darkness hypothesis". We prefer our more succinct sentence.

Line 114 – this is really far-fetched – a different species and totally anecdotal; it does not much strengthen the story

We believe this observation of a mallard colliding with local low-visibility obstructions is valuable for highlighting to readers the potential risks of (prey species') low-light flight in the area. Given the other reviewers did not take issue we would rather leave the observation in.

Line 116 – why would FIDs increase with birds less able to see the experimenter? They would then decrease?

Apologies if the wording is not clear. This is the "default" sensory limitation theory. i.e. under low light the birds simply can't see anything until it occupies a large portion of their visual field. So low light is predicted to yield lower FIDs.

Reviewer #2 (Remarks to the Author):

Brief summary of the manuscript

This manuscript aims to investigate whether artificial light at night (ALAN) affects prey behaviour (in this case Eurasian curlew). A range of hypotheses are presented for how birds may respond to light (or lack of) at night; sensory limitation, flight avoidance, cloaked in darkness (perceived safer) or fear of darkness (perceived riskier). Flight initiation distance in Curlew increased with ALAN, in line with the flight avoidance hypothesis (although other hypotheses cannot be entirely ruled out).

Overall impression of the work

Understanding the effects of anthropogenic effects such as light and noise is a very important area of research given the problems many wild animal populations face. Much has been done with noise in birds, but less with light, so I think this manuscript presents an original contribution to the field. Furthermore, I liked that it was especially grounded in theory, with good consideration of a range of hypotheses, well-linked to the established literature.

We are glad that the referee acknowledges the importance, and the lack of knowledge of the anthropogenic changes in nocturnal light environment on many animal populations, and finds our manuscript to be an original contribution to the field. We are also pleased that the referee supports our wide, theory-based approach in the manuscript.

I thought the standard of writing was generally high, with a clear narrative, that made it easy to understand what was done and why. My main criticism of the narrative is the lack of conservation emphasis. I appreciate there is a tight word limit and it is important to cover the behavioural theory but I do think it needs to be considered and emphasised more what the challenges are for shorebirds – there is a lot of conflict with development and recreation along coasts (of which ALAN is part of) and the Eurasian Curlew is particularly interesting since it has massively declined in the UK and is likely to go extinct in some areas (e.g. Wales). I think where it is argued shorebirds are a good model system in the introduction you could also introduce the conservation context without distracting from the behavioural narrative too much. I don't think your study needs to prove that ALAN is "bad" (indeed your study suggests it's more complex than that!) to be able to set things in a conservation context and attract a wide readership.

We thank the referee for the important perspective and argument to add into our work. We have further emphasised the importance of understanding the anthropogenic effects on curlew conservation in lines 59-63.

The study design is generally good. It is not experimental so there are potential confounds to be considered (similar to research along urban-rural gradients) e.g. other variables besides light could vary across the study site. These could perhaps be considered a bit more – there is consideration of environmental confounds such as

temperature but less discussion of what might vary across the site along the same gradient as light (i.e. other anthropogenic effects like disturbance or noise).

Human disturbance has been shown to affect FIDs of birds, however, the human activity in this area was very minimal at night, as the mudflat was difficult to enter (and did not interest people at night). In the daytime, few people were occasionally walking on the dry parts of the mudflats, while the curlew were foraging further away from them. If the curlew were used to human activity, one might predict shorter FIDs in daylight. However, we observed significantly longer FIDs in daylight compared to measurements taken at night. Additionally, the subset of data from the creek revealed the same correlations with light as the wider dataset, implying noise or other anthropogenic disturbance were unlikely to explain the effects given the creek was far more sheltered. Finally, we did not observe any change in FIDs during the season, thus arguing against the curlew's habituation.

However, I do think we need a combination of observational field studies such as this and smaller scale experiments (it would be quite hard to experimentally manipulate this particular study system), so I take no major issues with having an observational study per se.

We agree that studies in real-world systems are lacking in the ALAN literature, and believe we have made a useful contribution.

I am not an expert at measuring light, but the methods here seemed generally detailed and robust. What was less clear was whether there were likely to be repeated measures on the same Curlew (i.e. potential for pseudoreplication). I appreciate that the birds were unlikely colour ringed (or possible to read rings of anyway at night!) but some estimate of the sample size of the population size and therefore the likelihood of re-sampling would be useful for transparency over pseudoreplication possibilities (even if there is no data for this particular bay, there should be populations within the literature that give an estimate of general population size/density).

The highest observed number of curlews on the mudflats at once was around 50 individuals, so we estimated the curlew's population size to be around 50-70 individuals, as some individuals were foraging on the nearby fields. With our relatively small population size but large sample size, some repeat measures of individuals is inevitable, and very difficult to control in an unmarked population (even if they were marked, these marks would not have been visible at night). When testing the curlew individuals foraging in flocks we tried to avoid pseudoreplication by randomly selecting the approached individual (i.e., we did not systematically select the closest individual etc.). We also tried to test individuals widely across the study site (see the curlew locations in Fig S1.), and included the coordinates in the full model to control for possible individual preference on foraging locations. We have made these limitations more explicit in lines 270-276.

Finally, I wasn't too sure whether there is potential for observer bias since the observer was estimating the birds location on GPS – how accurate is this generally? I appreciate that using a laser rangefinder might be difficult at night and many studies likely use an estimation technique such as this, but did you measure your accuracy and check it didn't vary according to light levels? I would assume that I would be worse at estimating distance in the dark versus in the light, particularly in an environment where there aren't very many landmarks to relate a birds position to in a coastal context. Since most FID studies are done in daylight and don't look at a light gradient, I think that we can't necessarily assume the same accuracy using similar methods and it needs a bit more justification.

GPS location was found to be very accurate on this study site, as the mudflats did not have any obstacles weakening the GPS signal (e.g., vegetation, topographic variation). The GPS was turned on prior to fieldwork and left until its accuracy reached its highest level, then it was left on throughout fieldwork (improving accuracy). Also note that we did not conduct fieldwork in the rain (water reduces GPS signal). Absolute accuracy in GPS may be affected by satellite coverage and atmospheric, however the relative accuracy between two readings taken in short succession tends to be very high (less than 1m). Importantly, we found that ALAN levels (rather than natural light, which might confound with atmospheric and thereby GPS signal) explained FID best across the site as a whole, but not in the darker "creek" area. Taken together this implies that systematic error in GPS accuracy could not have explained our findings. We did experiment with using a laser rangefinder, but without built-in night vision it was impossible to use with such small objects as the curlew, and even just a minimal aiming error caused a significant measurement error on a topographic flat area. Please, see also our response to reviewer 1 about the FID measurement accuracy (visual landmarks and terminal footprints).

Overall though, I expect most of these comments can be addressed with minor revisions to the manuscript and I think this would be a valuable contribution to the published literature.

We thank the referee for positive feedback. We are glad that the referee considers our work as a valuable contribution to the previous work.

Specific comments

Lines 67-89. I thought there were some details here that were more appropriate to have in the methods or results section (e.g. references to figures and tables on line 73, detail about modelling from lines 84-89, reference to R code on line 89). I would cut these, as they repeat info in other sections and that would create more word count that could be used to introduce context on conservation status of Curlew.

Revised as suggested – reference to figures and tables removed.

Line 217. What about high winds? Wind speed can affect vigilance as the wind acts to mask noise so this could interact with FID.

Indeed, thank you for the relevant note. Line (214) added “The fieldwork was not conducted on windy nights to control for any acoustic cues of the observer and possible avoidance of taking flight in heavy wind.”

Line 220. I found it unclear here how you obtained weather data. The website you link to is just a weather forecast website. A weather forecast is different to measuring actual temperature at your field site or getting data from a nearby weather station.

Temperature was obtained from the hourly weather forecast. We did not have available local weather station data for temperatures. Revised as suggested: “weather data obtained from “<https://www.foreca.com/102640413/Penryn-Cornwall-United-Kingdom>”.

Line 222. www.bbc.com is a very vague web address – it’s just the entire bbc site. Provide a specific link for where you got the information from.

Revised as suggested: specific link provided.

Line 241. See point above – minimal on how bird location was recorded and the accuracy of this and whether light affects accuracy of estimation.

Please, see our response above, and our response to reviewer 1.

Line 258. I assume that horizontal light was measured at “bird height” not chest height of the observer? Needs to be stated what height above ground the horizontal measures were taken.

Revised as suggested: “at 40 cm height”.

Line 273. Related to my point on line 220, it’s unclear whether you actually measured temperature and how or whether you were using weather forecast data.

Please, see our response above.

Line 437. I liked that you reported your full model and that the results are similar whether you use a step-wise approach or full model – gave me extra confidence that the effect is large enough to be robust to what statistical methods you use.

We are glad that the reviewer found the presentation of our statistics clear. Indeed we were also delighted to have such robust findings from a behavioural ecology dataset with a number of potential confounds.

Reviewer #3 (Remarks to the Author):

In this article, the authors assess the FID of the Eurasian curlew in relation to light levels (moon and artificial light). They use FID to humans as a proxy for the fear felt by the birds. I like reading the paper and I think that the scientific question is interesting and timely, but I felt that it was too pretentious.

Given this study was conducted with a single species, it is too pretentious to say that artificial light affects the landscape of fear in shorebirds (just one studied!).

We have altered the title - see also our responses to reviewer 1.

Furthermore, and more importantly, the study area is quite limited in extension (around 600 m long). So, I bet that the number of curlews is also limited. Given that wintering population size is not given, I assume the authors have sampled many birds multiple times.

See our response to referee 2 regarding repeat sampling of individuals, and our associated edits.

Also some clusters of focal birds are observed in Fig S1. Can they be the same individuals? This is essential to answer such a question, as it affects statistical analyses (random factors) and to extrapolate to all shorebirds as the title proposes.

The higher density of observations in the narrow creek consists of measurements taken from both individual birds foraging alone and randomly selected individuals foraging in flocks of various size (ca. 5-35 birds). As we mentioned in our response to referee 2, it is likely that we re-tested some of the individuals during the study, however, we tried to avoid this by testing both individuals foraging alone and in the flocks of conspecifics by randomly selecting one individual for the experiment. As the curlew individuals were not ringed and the observation of rings was not possible at night, we were unable to identify individuals. However, we included the coordinates in the full model to control for possible individual preference on foraging locations.

I also miss a wider discussion on how artificial light can modify the landscape of fear of other birds (for example, other non-prey birds such as birds of prey or owls).

We are excited to explore the potential for the landscape of fear being affected in other species/systems, however we are also keen to avoid undue speculation at this stage, as there is simply a very limited amount of work in this area. E.g. reviewer 1 requested we exclude our observation of the other prey species, the mallard, colliding with the rigging of a boat, as being out of scope. However, we discuss this on lines 190–193: "Our results suggest ALAN may benefit the waders in reducing the costs associated with taking flight and could be predicted to reduce the optimal light conditions for the predator approaching the prey (naturally determined by the moon phases; Ferguson et al., 1987)".

FID studies and the landscape of fear are not the same. It is true that as a first approximation it could be useful, but a better description should be done.

We agree with the referee that FID and "landscape of fear" are not the same. In philosophical terms, we can never deduce the intentional state of another animal without language, so we must use inference based on observations. Existing "landscape of fear" studies use data such as animal movement, diet, and population dynamics to test for inferred fear. FID is a standard assay for perceived predation risk, and we have shown that FID is altered based on properties of the landscape. Moreover, the word "fear" is valuable to our arguments because it can encompass perceived predation risk with other risk factors associated with potential for severe injury or death that have rarely been considered (e.g. collisions). We have added a discussion of this in lines 65-68.

Line 215: what water level is this one? maximum tidal level? what is the tidal variation in depth?

Water level varies on site significantly, with a range of roughly 0.2 m – 6 m during large spring tides. The water level of 2.5 m that is mentioned in the Methods, corresponds approximately the middle to high tide mark, and was chosen based on mudflat topography, as enough mud is exposed at this water level to attract waders. Additionally, entering the study area is possible when the water level drops below this line, enabling the

fieldwork.

Revised as suggested: Line 205: "water level below 2.5 m, tidal variation on site between ca 0.2 m – 6 m"

Line 216: It must be reworded. Do you know if they are 86 and 27 different individuals? Are they ringed? How did you identify them?

See our response to referee #2. Our edits also make it clear that the curlew population was unmarked (line 270-275).

Lines 266: Are predictors standardized prior to modeling?

Yes, this is mentioned on lines 280-282: "Measurements of direct ALAN and downwelling light were log-transformed to meet model assumptions and all the other explanatory variables were standardised by centring and scaling, using the scale-function in R."

REVIEWERS' COMMENTS:

Reviewer #1 (Remarks to the Author):

Reviewer comments to the manuscript: 'Artificial lighting affects the landscape of fear in a widely distributed shorebird' by Juho Jolkkonen, Kevin J. Gaston and Jolyon Troscianko

The authors provide a revised manuscript of their study on the Flight Initiation Distance (FID) of curlews foraging on mudflats in the Penryn River, and how artificial light at night affects this parameter. The FID substantially increases with increased light levels, which effectively changes the 'landscape of fear' and thereby potentially foraging efficiency.

To my opinion, authors have sufficiently addressed all comments, and I do not have any further comments; I continue to like this work and look forward to see it published.